# Coagulopathies in Intensive Care Medicine: Balancing Act between Thrombosis and Bleeding

**DOI:** 10.3390/jcm10225369

**Published:** 2021-11-18

**Authors:** Friederike S. Neuenfeldt, Markus A. Weigand, Dania Fischer

**Affiliations:** Department of Anaesthesiology, Heidelberg University Hospital, 69120 Heidelberg, Germany; Friederike.Neuenfeldt@med.uni-heidelberg.de (F.S.N.); Markus.weigand@med.uni-heidelberg.de (M.A.W.)

**Keywords:** coagulopathy, coagulation, bleeding, critical care

## Abstract

Patient Blood Management advocates an individualized treatment approach, tailored to each patient’s needs, in order to reduce unnecessary exposure to allogeneic blood products. The optimization of hemostasis and minimization of blood loss is of high importance when it comes to critical care patients, as coagulopathies are a common phenomenon among them and may significantly impact morbidity and mortality. Treating coagulopathies is complex as thrombotic and hemorrhagic conditions may coexist and the medications at hand to modulate hemostasis can be powerful. The cornerstones of coagulation management are an appropriate patient evaluation, including the individual risk of bleeding weighed against the risk of thrombosis, a proper diagnostic work-up of the coagulopathy’s etiology, treatment with targeted therapies, and transfusion of blood product components when clinically indicated in a goal-directed manner. In this article, we will outline various reasons for coagulopathy in critical care patients to highlight the aspects that need special consideration. The treatment options outlined in this article include anticoagulation, anticoagulant reversal, clotting factor concentrates, antifibrinolytic agents, desmopressin, fresh frozen plasma, and platelets. This article outlines concepts with the aim of the minimization of complications associated with coagulopathies in critically ill patients. Hereditary coagulopathies will be omitted in this review.

## 1. Introduction

Coagulopathies are common among critically ill patients and associated complications may be preventable if anticipated and treated in a goal-directed, risk-adjusted manner at an early state. Patient Blood Management measures include the optimization of hemostasis to minimize blood loss, which may significantly impact morbidity and mortality. This article will outline both thrombotic and hemorrhagic conditions as they may coexist and the overtreatment of one condition may cause the other. The cornerstones of coagulation management are an appropriate patient evaluation, including the individual risk of bleeding weighed against the risk of thrombosis, a proper diagnostic work-up of the coagulopathy’s etiology, treatment with targeted therapies, and appropriate transfusion of blood product components. 

## 2. VTE and Bleeding Risk Assessment Tools

In health, hemostasis is a complex, tightly regulated balance between bleeding and clotting, which is often deranged in critical care patients for multifactorial reasons, sometimes even in a way that thrombotic and hemorrhagic conditions can exist simultaneously [1]. The risk of bleeding and thromboembolic complications vary greatly depending on the patient population, grade of organ dysfunctions, nature and severity of disease, pre-existing conditions, necessity, and scope of medical interventions. This article outlines these risk factors as well as concepts with the aim of the minimization of complications associated with coagulopathies by evidence-based measures in critically ill patients. 

Thrombotic events are common in critically ill patients, both before and after intensive care unit (ICU) admittance, even in patients receiving routine thrombosis prophylaxis [2]. In particular, the development of pulmonary embolism or lower extremity deep vein thrombosis is associated with increased mortality. The risk of nonleg venous thrombosis is comparatively lower and not necessarily associated with higher mortality [3]. Risk scores for hospitalized medical patients, such as the Padua Prediction Score are available to approximate an individual patient’s risk for thrombosis [4]. However, applicability in high-risk critical care patients is limited, which makes patient assessment and clinical research on strategies to prevent or stop thromboembolic complications difficult [5]. Specifically for ICU patients, the ICU-venous thromboembolism score was developed, including six independent predictors: Central venous catheterization, immobilization greater than or equal to 4 days, prior history of venous thromboembolism, mechanical ventilation, lowest hemoglobin during hospitalization greater than or equal to 9 g/dL, and platelet count at admission greater than 250,000/μL [6]. Further risk factors identified in a meta-analysis of observational studies include older age, higher body mass index, active malignancy, history of recent surgery, sepsis, lack of pharmacologic venous thromboembolism prophylaxis, and the use of vasoactive medications [7]. Thrombosis prophylaxis reduces the incidence of VTE and the guidelines recommend pharmacologic prophylaxis for all of the critically ill patients, if not contraindicated [8,9,10]. It is important to consider that most of the patients with cancer face a 4- to 7-fold higher risk of developing VTE compared to patients without cancer, due to an increased expression of tissue factor by malignant cells [11,12]. Moreover, patients with COVID-19 require special consideration as COVID-19 often manifests in complications, such as deep vein thrombosis, pulmonary embolism, arterial thrombotic events, and disseminated intravascular coagulopathy, which will be discussed in an extra paragraph below.

The most severe complication of anticoagulation is major bleeding. Before anticoagulation begins, absolute contraindications need to be ruled out, including ongoing intracranial or other life-threatening bleeding, recent major surgical trauma, and thrombocytopenia <30 × 10^9^/L [13]. In patients with elevated PTT for unknown reasons, further evaluations and testing are warranted before anticoagulation is considered as inherited or acquired causes may be present, such as von Willebrand syndrome or factor deficiencies. Additionally, modifiable risk factors for bleeding should be identified and optimized. Of note, the timeframe between a surgical procedure and the “safe” initiation of anticoagulant treatment may vary and should be defined at the individual level. Although the use of pneumatic compression devices for VTE prophylaxis poses seemingly little harm, their efficacy has not been proven yet and they may be associated with additional resource use and cost, implying that critically ill patients would rather be treated with pharmacologic prophylaxis as soon as it is safe [13,14,15].

The risk of VTE and the decision to use mechanical or pharmacologic prophylaxis needs to be weighed against the risk of bleeding as bleeding prevalence, especially among surgical ICU patients, is high and associated with a higher risk of in-hospital morbidity and mortality [16,17]. The “Bleeding score” by Decousus et al. identifies risk factors at admission associated with in-hospital bleeding risk in acutely ill medical patients [18].

Further risk factors for bleeding can be exemplified by the HEMORR_2_HAGES score, which is generally used to stratify patients’ risk for bleeding after anticoagulation for atrial fibrillation in synopsis with situation specific risks and benefits [19]. The criteria added to the risk of bleeding are displayed in Table 1. 

Further potential reasons for bleeding include acquired or hereditary bleeding disorders (platelet function abnormalities, factor deficiencies, and factor inhibitors), hepatic or renal dysfunctions, renal replacement therapy, recent surgery, concomitant anticoagulation medications, and further drugs that may affect coagulation such as cephalosporins, ginkgo preparations, interferon, selective serotonin reuptake inhibitors or tricyclic antidepressants [20]. The management of bleeding is based on the understanding of its contributing factors. In addition, treatment measures should be chosen with care and closely monitored for efficacy and side effects in order to maintain the balance.

## 3. Management of Coagulopathies

A prerequisite in the management of coagulopathies is a multidimensional patient assessment. The evaluation of the clinical situation as well as laboratory and point-of-care based analysis will be outlined below, with a focus on inherited disorders. Furthermore, anticoagulant reversal and treatment of specific bleeding situations that an intensivist will be confronted with will be outlined. Irrespective of other reasons for coagulopathy, the essential and indisputable preconditions of hemostasis are pH > 7.25, ionized calcium >1 mmol/L, and body temperature > 34 °C [21].

Generally, any bleeding patient should be risk-stratified based on hemodynamic instability, timing of the last dose of anticoagulant agent, source and quantity of blood loss [22]. The pattern of bleeding, which ranges from petechiae to mucosal bleeding, and generalized oozing from de-epithelialized surfaces as well as fast bleeding from major vessels, determines whether interventional, surgical, mechanical options or topical hemostatic agents or drug therapy may be indicated in the first line. The scope of therapeutic reactions ranges from cases of minor bleeding, for instance, in anticoagulated patients, where it may be reasonable to simply pause the anticoagulant and closely monitor the development of the bleeding, hemodynamics, and volume status. In cases of major bleeding, depending on physiological factors and the complexity of an injury, an urgent damage control surgery may be necessary as a first line approach in order for the patient to be stabilized at the intensive care unit first. Then, a definite surgical repair may be applied [23].

### 3.1. Laboratory and Point-of-Care Assessment of Coagulation Disorders

The earlier a coagulopathy is diagnosed, the earlier a targeted therapy can be initiated. Nevertheless, laboratory abnormalities are not to be corrected with blood products unless there is a clinical bleeding problem, a surgical procedure is required, or both. An exception may be platelet counts, which will be specified below. Generally, it is important to consider that many different conditions can produce similar laboratory abnormalities, highlighting the importance of anamnesis, physical examination, and clinical judgement. Standard coagulation tests include prothrombin time (PT), a test of the extrinsic coagulation pathway and activated partial thromboplastin time (aPTT), as well as a test of the intrinsic coagulation pathway. Both coagulation tests were primarily designed to monitor anticoagulants, such as warfarin and heparin. They do not correlate with the risk of bleeding or thrombosis. These tests only evaluate the time to the start of clot formation, which is only a small window of the coagulation cascade. Platelet count, fibrinogen and d-dimer levels, bleeding time, and findings on blood smear provide further information. However, none of these standard coagulation tests reflect the balance between the actions of pro- and anticoagulant factors [24]. On the other hand, viscoelastic testing provides a more global picture of hemostatic function as it allows for the measurement of the interaction between the humoral coagulation pathways with platelets, monocytes, and fibroblasts. This is exemplified by the fact that hypocoagulability in septic patients may be seen only in impaired thromboelastography, whilst standard coagulation tests such as PT and aPTT fail to reflect it [25]. Viscoelastic testing appears favorable in reducing blood product transfusions, “bleed-to-treat” time or “turnaround time”, especially in cardiac surgery patients [26]. While the results are promising, no systemic larger trials in critical care have taken place to date and prospective randomized trials are needed with respect to clinical outcomes for critically ill patients. Point-of-care devices continuously evolve. Real-time testing of anti-Xa activity, international normalized ratio (INR), as well as detection of direct oral anticoagulants (DOAC) have become more reliable and widely available, which can be very helpful in guiding the prevention of thromboembolic events or the need for reversal [26,27,28]. 

When an urgent reversal of anticoagulation is required, potent recombinant coagulation products and innovative reversal agents are available or under investigation. Table 2 summarizes the reversal agents of various anticoagulants [29]. However, in particular, the reversal of DOACs with specific antidotes requires high vigilance and careful dosing as reversal agents may develop tremendous thrombogenic potential, especially in combination with other hemostatic drugs. Robust evidence from clinical trials is needed to demonstrate their beneficial potential and provide further guidance for their handling in critical care medicine.

In vitamin-K-treated patients, the administration of four-factor *PCC* to reverse vitamin-K anticoagulant effects is indicated [30]. If PCC is not available, then in bleeding patients where vitamin-K-induced coagulopathy is considered a contributing factor, the transfusion of plasma (15 to 20 mL/kg) in addition to 5 to 10 mg IV vitamin K is recommended.

Platelet aggregometry provides additional information on platelet function and show the effects of antiplatelet drugs. Drugs that interfere with platelet function include aspirin, clopidogrel, prasugrel, dipyridamole, and the glycoprotein IIb/IIIa (GP IIb/IIIa) inhibitors. Irreversible inhibitors bind covalently to the platelet receptor, thus deactivating it permanently. Consequently, the platelet is unable to activate adenosine diphosphate for the whole life span of platelets ranging from 7 to 10 days, in order for the substitution to possibly be necessary despite quantitatively normal platelet levels. Furthermore, it is important to consider drug half-lives and drug levels that remain in the circulation, which will also affect allogeneic platelets after transfusion. 

Thrombocytopenia is common in critically ill and surgical patients and there is a wide variety of platelet transfusion practices [31]. In the evaluation of thrombocytopenia in ICU patients, the possibility of pseudothrombocytopenia is important to consider. It is an ex vivo thrombocytopenia, which can be ruled out by direct microscopic examination of a well stained blood smear from EDTA-venous blood or repeated blood counts in citrated or heparinized blood. If thrombocytopenia is confirmed, immune-mediated processes should be thought of as possible reasons, especially for rapid decreases in platelet counts. Immune-mediated processes lead to increased destruction of platelets either by autoantibodies in immune thrombocytopenia (ITP) or drug-dependent antibodies (D-ITP) or alloantibodies in post-transfusion purpura [32]. An important differential diagnosis is heparin-induced thrombocytopenia (HIT), which is the commonest cause of immune-mediated coagulation disorders and may occur in the course of heparin exposure. However, in ICU patients, many other confounding factors can also cause thrombocytopenia. In addition, widely available screening tests for HIT, such as detection of platelet factor 4 antibodies on enzyme-linked immunosorbent assay, show a high rate of false positives in up to 80% of patients, which is the reason that a confirmatory test (e.g., serotonin release assay) should be performed if the initial screening test is positive [33]. In patients with low platelet counts under stable conditions without antiplatelet therapy, a transfusion threshold of 10,000 per cubic millimeter is both hemostatically efficacious and cost-effective in reducing platelet-transfusion requirements [34]. However, the data founding this recommendation mainly stem from medical patients. In surgical patients, it can be argued that higher safety thresholds should be maintained, especially if the patient is actively bleeding or has increased platelet turnover. Here, a platelet count over 50,000 per cubic millimeter is advised [35].

### 3.2. Disseminated Intravascular Coagulation

Sepsis is the most common cause of disseminated intravascular coagulation (DIC) [36]. This acquired syndrome is characterized by a dysfunctional systemic activation of coagulation pathways leading to microvascular thrombosis and subsequent depletion of coagulation factors and platelets [37]. DIC may result in widespread thrombosis, multi-organ failure, and profound hemorrhage from various sites [38]. For the clinicopathological diagnosis of DIC, the scoring system of the International Society on Thrombosis and Hemostasis (SIC-Score) shown in Table 3 can be used [39]. A score of 4 or more is defined as sepsis-induced coagulopathy, which correlates with a mortality rate greater than 20% and indicates the initiation of anticoagulation therapy [40]. Prolonged prothrombin time as well as levels of D-dimer and fibrin degradation products may provide additional information. 

Patients with hematological and solid cancers also have an increased risk of DIC [41]. In contrast to sepsis-induced DIC, where organ failure is predominantly caused by insufficient fibrinolysis, cancer-induced DIC is often characterized by distinctive hyperfibrinolysis [42].

The basic principle in managing DIC is always the treatment of the underlying disorder. Further management may not be necessary and is advised against in patients with mild abnormalities in coagulation and no evidence of bleeding [43]. Guidelines suggest transfusion to maintain a platelet level of more than 50,000 / mL^3^ and fresh-frozen plasma to maintain a prothrombin time and aPTT of less than 150 % the normal control time, as well a fibrinogen level of more than 1.5 g per liter [44,45]. Antifibrinolytic agents are contraindicated in the management of DIC, as the fibrinolytic system needs fibrin during the recovery phase to ensure the dissolution of widespread. Other possible contributing causes should be considered. In patients with sepsis-induced coagulopathy, the activity of the endogenous anticoagulant ATIII is reduced. To date, prospective trials on supplementation therapy in these cases have not been able to show a significant reduction in mortality [46]. Currently, the treatment with ATIII is only approved and commonly used in Japan [47]. Another coagulant inhibitor is recombinant soluble thrombomodulin (rTM). While both the retrospective and prospective studies, such as the SCARLET trial, showed beneficial effects on patients with sepsis, no significant risk reduction in mortality has been demonstrated for rTM in comparison to the placebo [48,49]. 

Despite multiple randomized, controlled trials, anticoagulant therapy for patients with sepsis and DIC is controversial and no globally agreed upon therapy regime exists. Studies have shown the potential of anticoagulant therapy to improve outcomes and reduce mortality in patients with sepsis-induced coagulopathy, underlining the importance of early identification of sepsis-induced coagulopathy [50]. Currently, there are no recommendations as to which type of heparin might be the most effective. A comparison of studies between the low-molecular-weight heparin and unfractionated heparin, to this date, failed to demonstrate a significant difference in mortality [51]. Moreover, antiplatelet therapy has been investigated in septic patients. The ANTISEPSIS study including 16,703 patients could not confirm the earlier observational data suggesting a lowering of aspirin on the in-hospital mortality in sepsis [52,53]. A recent meta-analysis including 10 studies with a total of 36,514 patients showed that a treatment with ticagrelor correlates significantly with a reduced incidence of pneumonia, although not with other infections or sepsis [54]. As a potent P2Y12 inhibitor, ticagrelor is able to reduce the intensity of an inflammatory response as shown in the experimental settings, but further data are needed for a conclusive assessment [36].

### 3.3. Massive Hemorrhage and Trauma-Induced Coagulopathy

Significant hemorrhage may occur in patients who are critically ill following surgical or traumatic injury. Patients may also hit the ICU with existing dilutional or trauma-induced coagulopathies following prehospital or intraoperative volume replacement and/or transfusion of allogeneic blood products. In these patients, it is important to consider that both cellular and humoral factors of coagulation need replacement in concentrations and constellations, which are heavily dependent on the etiology of coagulopathy and the resuscitative measures undertaken before ICU admission [55]. Therefore, in the critical care setting, a fixed ratio of red blood cell (RBC) to plasma (fresh frozen plasma, FFP) and to platelets most likely does not address the very heterogenous coagulopathies encompassed here. Rather, viscoelastic and aggregometric coagulation monitoring as well as goal-directed coagulation management are probably more appropriate, although conclusive studies in critical care are missing [56,57,58]. It is always important to strive for or maintain optimal hemostatic conditions with normal pH, temperature, and calcium levels. 

The European Society of Anesthesia (ESA) recommends the early and targeted treatment of coagulation factor deficiencies with coagulation factor concentrates in perioperative bleeding management [30]. A few studies demonstrate an advantage for a clotting factor based therapy in massive bleeding, as the targeted administration of coagulation factor concentrates was more effective in the correction of trauma-induced coagulopathy than the transfusion of FFP [59,60,61]. However, a protective effect of FFP after hemorrhagic shock is postulated, which may go beyond the clotting effects and may be explained by a stabilization of the endothelial cell permeability and integrity [62]. In a mouse model of trauma and hemorrhagic shock, FFP effectively reduced vascular hyperpermeability and inflammation [63]. Moreover, in vitro models of vascular endothelial cells demonstrate protective effects of FFPs, as the FFP inhibits permeability, endothelial adherens junction breakdown, and endothelial white blood cell binding [64]. Furthermore, findings from a hemorrhagic shock model in rats support the concept of cardiovascular and microvascular stabilization by infused FFP, as an increase in microvascular perfusion associated with restored endothelial glycocalyx could be demonstrated [65]. FFP may also attenuate the inflammatory response of endothelial cells with regards to neutrophil-endothelial interactions [66]. Nevertheless, in terms of coagulation management, it is important to consider that although the plasma contains all of the clotting factors, transfusion of FFP in bleeding patients does not achieve the sustained correction of coagulation [67]. PCC, directed at specific phases of coagulation which are identified by alternative laboratory assays, offers the advantage of smaller volumes of resuscitative fluids. In dynamically changing situations, turnover rates and half-lives are important for consideration when dosing coagulation therapy. In severe trauma and massive hemorrhage, it is crucial to consider that plasma fibrinogen levels decrease early and significantly faster than the other coagulation factors. As fibrinogen is highly critical in hemostasis and clot formation and rapidly depleted, fibrinogen administration through fibrinogen concentrates is an important treatment component [68]. Platelet andanti-fibrinolytic agents, such as tranexamic acid, as well as the use of recombinant activated factor VII in selected cases of refractory to standard treatment also need to be taken into consideration [69,70]. In particular, the early use of tranexamic acid safely reduces the risk of death in bleeding trauma patients and in post-partum hemorrhage [71]. 

Whether volume replacement and endothelial membrane stabilization through plasma transfusion outweigh the advantages of factor concentrates, is probably very context-sensitive. In addition, larger-scale prospective trials in humans with a focus on microcirculation are required. 

### 3.4. Liver Disease

Most of the hemostatic proteins (coagulation factors and physiologic anticoagulants) are synthesized in the liver. Furthermore, the liver is a site of the metabolism of sialic acid residues from fibrinogen, activated coagulation factors, and tissue plasminogen activator. In chronic liver disease, the decreased synthesis of coagulation factors combined with a reduced production of physiologic anticoagulants (C and S proteins) and the fibrinolytic system, generally lead to a state of “rebalanced coagulation” [72,73]. Standard laboratory tests, such as prothrombin time or INR, are not equipped to reflect this “new” balance. In addition, pathological findings do not translate into bleeding risk or the need for coagulation factors [74,75,76]. Viscoelastic tests may be superior in evaluating clot formation ability in patients with liver disease [77,78]. Coagulation therapy should be considered for patients at high risk of bleeding, with a scheduled invasive procedure or for patients actively bleeding [79,80]. Regarding the role of prophylactic FFP therapy prior to central venous catheterization, Rocha et al. compared three transfusion protocols in critically ill cirrhosis patients in a randomized controlled approach, including 57 patients [81]. No difference in bleeding was found. In this relatively small cohort, the restrictive strategy significantly reduced blood transfusion and costs in patients with cirrhosis. However, the study cohort may simply have been too small to find significant differences [82]. Nevertheless, this underlines the importance of ultrasound guidance for central venous catheter placement in patients at risk for bleeding, which is probably increasingly effective and cheaper in the prevention of bleeding than any preprocedural correction of hemostasis. 

In patients with liver disease and laboratory tests indicating the abnormal synthesis of vitamin-K-dependent coagulation factors, vitamin K should be routinely administered to aid in the PCC. On the other hand, it should only be used with the greatest care as it only contains pro-coagulant factors, shifting the balance towards thrombosis. FFPs may be a safer option, but pose a risk of transfusion-related circulatory overload (TACO), as even high doses of plasma only result in a moderate increase of clotting factor and inhibitor activities in the recipient [83]. In bleeding patients with liver dysfunction and PT below 50%, the plasma could be transfused at a dose of 20 mL/kg body weight [84]. The objective of the treatment is to arrest bleeding and to increase PT to at least 50%. The use of FFP in the management of portal hypertensive bleeding or preoperative prophylaxis is not recommended [85,86]. It should be considered that the transfusion of blood products may also increase the risk of further bleeding or recurrent bleeding in patients with cirrhosis due to the increasing portal pressures and altering coagulation parameters [87]. Patients with severe cirrhosis often show decreased levels of fibrinogen, as well as impaired functionality of fibrinogen, which is called dysfibrinogenemia [88,89]. Recent studies have found low fibrinogen to be an independent risk factor for increased bleeding, as well as a predictor of mortality [88,90]. Although clinical trials have yet to define a sufficient fibrinogen level, maintaining fibrinogen levels above 100–120 mg/dL during blood loss have been suggested based on experts’ opinion [91,92].

In liver disease, thrombocytopenia is often present and caused by, for example, decreased thrombopoietin production, hypersplenism, and platelet activation [87,93,94]. Although there is no evidence-based transfusion threshold, based on expert opinion, a platelet transfusion in patients with cirrhosis is generally recommended to a threshold of 50 × 10^9^/L in the presence of bleeding or invasive procedure [95]. Although it should be kept in mind that the platelet quantity does not necessarily correlate with the platelet function. This explains why no clinical study has been able to determine a reliable platelet threshold to prevent bleeding [87]. Consequently, every platelet transfusion requires an individual risk-benefit analysis.

In particular, cirrhotic patients are simultaneously at high risk for portal vein thrombosis (PVT). Its prevalence ranges from 10 to 25% depending on the Child-Pugh score [96]. Due to its aggravating effects on cirrhosis, fibrosis, and variceal bleeding, a sufficient PVT treatment is pivotal [96,97,98]. As a general rule, the dose and timing of anticoagulant prophylaxis around surgical procedures or other high bleeding risk intervals should be carefully chosen. 

### 3.5. Renal Disease 

In patients with kidney insufficiency, the accumulation of uremic toxins may result in platelet dysfunction, which typically presents with ecchymoses, purpura, epistaxis, and bleeding from puncture sites [35]. Dialysis, cryoprecipitate, desmopressin, and tranexamic acid improve uremic bleeding. Continuous dialysis is a common requirement in critical care patients with renal insufficiency. In addition, sufficient anticoagulation is necessary for the prevention of blood clotting, as premature clotting of the dialysis circuit leads to increased iatrogenic blood loss. However, anticoagulation itself inherently increases bleeding risk. The use of citrate-anticoagulation has proven a worthy therapeutic strategy that extends the filter lifetime without increasing the risk of bleeding during dialysis [99,100,101]. 

### 3.6. COVID-19 

COVID-19 is associated with thromboembolic events, which are explained by a complex interplay between the coronavirus, the coagulation system, endothelial cells, and the immune system’s response to infection [102]. In several stages of COVID-19, the activation of the coagulation system manifests in complications, such as deep vein thrombosis, pulmonary embolism, arterial thrombotic events, and disseminated intravascular coagulopathy. Results of a meta-analysis of 35 studies and 6427 patients demonstrate that a severe COVID-19 infection is associated with higher D-dimer values, lower platelet count, and prolonged PT. This data suggest a possible role of disseminated intravascular coagulation in the pathogenesis of severe COVID-19 infection [103]. A favorable outcome has been reported with the use of heparin in COVID-19 patients, especially in those with markedly high D-dimer levels or with sepsis-induced coagulopathy [104]. However, the beneficial effect of therapeutic anticoagulation is diminished and the risk of hemorrhage is increased in patients with progressively more severe disease, potentially related to hyperinflammation, endothelial disruption, platelet activation, and coagulopathy. The use of higher than standard prophylactic-dose anticoagulation is not beneficial, especially in ICU patients, as the incidence of bleeding rises without a benefit in terms of venous or arterial thromboembolism treatment, with extracorporeal membrane oxygenation or mortality within 30 days [105,106]. The role of antiplatelet therapy remains to be determined, as well as other therapies that might modulate the prothrombotic characteristics of SARS-CoV-2 in hospitalized patients.

### 3.7. Mechanical Circulatory Support Devices

Anticoagulation in the setting of mechanical circulatory support devices is another challenging aspect of critical care. Mechanical circulatory support devices, such as extracorporeal membrane oxygenation (ECMO), left ventricular assist devices (LVAD) or percutaneous ventricular devices, such as the Impella^®^, are associated with a coagulopathy characterized by both thromboembolic and hemorrhagic complications [107]. In most of the patients with an indication for these devices, plasmatic coagulation and platelet aggregation are already impaired even before the initiation of the assist device, due to systemic inflammation, organ failure, anticoagulants such as heparins, phenprocoumon, apixaban or antiplatelet medication. Contact of blood components with artificial surfaces, shear stress, and hemodilution add to humoral and cellular coagulation defects. Fibrinogen levels, fibrin polymerization, platelet activation, and release of extracellular vesicles may also be deranged depending on the form of assist device. In particular, during ECMO, abnormal platelet adhesion, delayed extrinsic pathway activation represented by an increased international normalized ratio, and mildly reduced Factor XIII activity may be major contributors to bleeding complications [108]. Furthermore, shear forces induce an acquired von Willebrand disease, contributing to hemorrhagic events. At the same time, contact activation of plasmatic coagulation and platelets at the artificial surfaces of the respective system may contribute to prothrombotic platelet activation, necessitating the routine systemic administration of anticoagulants for the prevention of thromboembolic occlusion of the mechanical circulatory support devices. Heparin is primarily used for anticoagulation, which may result in heparin-induced thrombocytopenia, which can further complicate ECMO- and LVAD-associated coagulation dysfunction. The systemic anticoagulation, again, adds to the risk of bleeding complications. The intensivist needs to maintain a fragile balance between bleeding and clotting. 

### 3.8. Acute Burn-Induced Coagulopathy 

Patients with severe burns frequently develop acute-burn-induced coagulopathy (ABIC), a complex process of uncontrolled coagulation and fibrinolysis [109]. The pathophysiology of ABIC remains largely unknown. Several studies suggest a multifactorial pathogenesis based on tissue hypoperfusion and hypoxic injuries, systemic inflammation, hypothermia, and dilutional coagulopathy following a large fluid resuscitation volume [110,111,112]. The extent of total body surface area, as well as the degree of tissue hypoxia are risk factors for the occurrence of ABIC [113]. Several retrospective studies showed a strong correlation of ABIC with burn-related mortality [110,113,114]. ABIC occurs after 4 to 16 h in approximately 30–40% in patients with severe burns [110,113,115]. Previous studies defined that the ABIC based on INR is higher by 1.2 or 1.3, but laboratory markers, such as INR and aPTT, turned out to be unreliable indicators of coagulation in patients with severe burns [116]. 

The optimal fluid therapy for patients with severe burns is highly complex. On the one hand, a prompt and sufficient shock treatment is crucial to avoid tissue hypoperfusion to trigger coagulopathy [110]. On other hand, the significant amounts of fluids needed may also cause a dilutional coagulopathy, as well as an impaired functionality of fibrinogen. In the absence of clear recommendations, the need for individual risk-benefit-analysis remains when treating burn patients [117]. Viscoelastic tests allow a more accurate and faster management of ABIC, although evidence-based recommendations are still lacking [116,118]. A retrospective study found that the intravenous administration of tranexamic acid in patients with severe burns reduced the red blood cell transfusions and the necessity for regrafting without an effect on mortality [119].

### 3.9. Role of Plasma in the Treatment of Coagulopathy

FFP transfusion in critically ill patients is associated with the increased risk of infection and transfusion-related circulatory overload, as well as acute lung injury [120,121,122,123]. Nevertheless, there is a substantial use of FFP in the ICU with wide treatment variability, suggesting the assumption that clinical indications go beyond coagulation management [124,125,126]. Among the ICU patients with abnormal INR, one third receives FFP transfusions, 50% of FFP transfusions are given to nonbleeding patients, and 40% to nonbleeding patients whose INR is either normal or only modestly deranged (≤2.5). The dose of FFP is also highly variable (median dose 10.8 mL/kg). As only moderate increases of the clotting factor and inhibitor activities can be expected in the recipient of plasma transfusion, a sufficient FFP dose and transfusion speed is required for any effective coagulation therapy (a minimum of 15 mL/kg body weight, infusion rate of 30–50 mL/min). In the average adult patient, any dose below 600 mL (2 to 3 units of FFP) is insufficient [83,127]. Additionally, the increased turnover of coagulation factors and inhibitors, due to the consumption and/or loss or dilution in patients with coagulopathies and bleeding, need to be considered when planning transfusion intervals [127]. However, in patients with an impaired heart, liver or renal function, the plasma dose is limited due to the risk of hypervolemia. PCC contains three to four clotting factors (3F: FII, FIX, FX; 4F-PCC: +FVII). On average, the concentration of clotting factors is approximately 25× of plasma. However, PCC concentrates do not contain fibrinogen, factor V, factor VIII, vWf, factor XI, and factor XIII. In addition, they are singularly used. Therefore, no replacement for the plasma is needed when treating complex coagulopathies.

The battle-deciding factor for an effective treatment of coagulopathy is not the sum of the coagulation factors that are substituted, but their concentration at the site where a thrombus needs to be built [128]. Therefore, the use of plasma transfusion for coagulation management is limited by volume, efficacy, and the individual risk of adverse transfusion reactions. The latter obviously depends on patients’ underlying thrombotic risk factors, dosing, and indication for plasma usage. Special risk groups for side effects of FFP transfusion or coagulation factor concentrates should preferentially be treated restrictively or with the respective alternative.

However, there might be further roles for FFP beyond the clotting therapy, for instance, when used in plasmapheresis for cytokine reduction [129]. In the frame of the COVID-19 pandemic, plasmapheresis has gained momentum and might be beneficial in special patient cohorts as an extracorporeal method for immunomodulation. 

Despite the common recommendations that FFP should not be used as a volume expander in the absence of coagulation deficiencies and active bleeding, FFP transfusion rates of up to 57% in sepsis patients suggest a high popularity [84,130,131,132,133]. This may be inadequate in terms of coagulation management, but there is a correlation between endothelial damage and coagulopathy in septic patients [134]. As FFP has diverse effects on the parameters of endothelial condition and inflammatory status, it might have the potential to benefit in terms of an enhanced microcirculation, hemodynamic stability, and vasopressor sparing effect [124]. Despite the ongoing controversy on the effect of FFP transfusion on systemic inflammation and endothelial damage, a positive effect on the endothelial integrity was postulated in observational studies and shown in experimental animal models [64,134,135,136,137]. A reduction of vascular hyperpermeability and inflammation was seen in vitro in pulmonary vascular endothelium [63].

Whether the restoration of endothelial integrity observed in the above mentioned experimental models may translate into improved microcirculation and consecutive vasopressor sparing effects or an even better outcome in septic patients, should be carefully weighed against the potential side effects of FFP in randomized controlled trials that evaluate restrictive vs. liberal FFP transfusion strategies. 

## 4. Conclusions

Coagulopathy is common in intensive care and is often multifactorial. Prognostic factors for the development of coagulopathy can be identified at the ICU admission. In addition, these factors may be used to plan anticoagulation and select patients at higher risk in future randomized clinical trials. Tailoring individualized concepts in order to minimize the complications associated with coagulopathies is highly challenging in critical care patients, especially as the medications that modulate hemostasis can be powerful. FFPs remain the broad-spectrum therapy for the correction of coagulopathy, especially in patients with active bleeding. However, they are also transfused frequently in patients with abnormal coagulation tests to prevent bleeding or for further non-evidence-based indications. Potential glycocalix-stabilizing and immunomodulatory effects are promising themes for future studies to determine the risk-benefit-ratio for FFP transfusion, especially in hemodynamically instable patients. It is crucial to find the underlying cause of coagulopathy and understand the limitations of various tests to assess them. Viscoelastic point-of-care testing alone or combined with platelet function testing provides prompt diagnosis of coagulopathy and allows for targeted treatments in bleeding patients. 

In conclusion, the management of coagulopathy is based on the understanding of its contributing factors. Patient Blood Management advocates careful patient risk assessment, prompt evaluation of coagulopathies, and implementation of goal-directed strategies that reduce thrombosis and bleeding in an individualized treatment approach in order to improve the patient outcome.

## Figures and Tables

**Table 1 jcm-10-05369-t001:** Risk factors for bleeding according to the HEMORR_2_HAGES score [19].

Criteria
Hepatic or renal disease
Ethanol abuse
Malignancy
Older (age > 75)
Reduced platelet count or function
Rebleeding (Prior Bleed)
Hypertension (uncontrolled)
Anemia
Genetic factors (CYP 2C9 single-nucleotide polymorphisms)
Excessive fall risk
Stroke

**Table 2 jcm-10-05369-t002:** Common anticoagulants, diagnostic tests, and their respective reversal agents [29]. Activated partial thromboplastin time (aPTT), prothrombin time (PT), international normalized ratio (INR), activated clotting time (ACT), prothrombin complex concentrate (PCC), fresh frozen plasma (FFP).

Anticoagulant	Laboratory Test	Reversal Agent
Aspirin Clopidogrel, ticagrelor Prasugrel	multiple platelet function analyzer	Consider the use of desmopressin Platelet transfusion
unfractionated Heparin	aPTT	Protamine
Vitamin-K-Antagonists	PT/INR	Vitamin K Prothrombin complex concentrate (PCC) (FFP)
Low molecular weight heparins (LMWH)	Anti-FXa	Protamine (partial) Aripazine
Fondaparinux	Anti-FXa	recombinant activated factor VII (partially) Aripazine Activated PCC Andexanet alfa
Factor Xa inhibitors (apixaban, rivaroxaban, edoxaban, betrixaban)	Anti-FXa	Andexanet alfa (irreversible) Aripazine PCC
Dabigatran	Limited value except Thrombin Time, Ecarin Clotting Time, TEG, anti-FIIa	Idarucizumab (irreversible) Aripazine PCC
Argatroban	Limited value except TEG Anti-FIIa PTT or ACT	No specific antidote
Alteplase	D-dimer	Tranexamic acid (partial)

**Table 3 jcm-10-05369-t003:** Sepsis-induced coagulopathy (SIC) score by the International Society on Thrombosis and Hemostasis [39].

SIC-Score		
Item	Range	Score
platelet count (−10^9^/L)	<100	2
	>100, ≤150	1
prothrombin time (PT ratio)	>1.4	2
	>1.2 ≤ 1.4	1
SOFA score	≥2	2
	1	1

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
