# Peer review of "Coagulopathies in Intensive Care Medicine: Balancing Act between Thrombosis and Bleeding"

_jcm, 2021, doi:10.3390/jcm10225369_

Round 1

Reviewer 1 Report

In their manuscript the authors reviewed some coagulopathies that are common in critically ill patients. Although the manuscript is potentially interesting, several issues arise.

Major comments

Abstract

The abstract is confused and the topic that will be treated is not clear (patient blood management in critically ill patients or coagulopathies or anticoagulation or reversal?). The authors seem to address the topic on patient blood management (BPM) and they point out that the cornerstones of coagulation management are patient evaluation for individual thrombosis and bleeding risks, analysis of different coagulopathies and their treatment with a goal-directed therapy. However, the optimization of hemostasis is not explained in the following text.

Introduction

The reader would expect the introduction to the topic coagulopathies or BPM. However, the evaluation of thrombotic and haemorrhagic risk in critically ill patient is explained. If the authors decide to discuss about thrombotic risk (Padua score has been create for hospitalized medical patients), I would suggest referring also a new score for critically ill patients (see Viarasilpa T, et al. Prediction of symptomatic venous thromboembolism in critically ill patients: the ICU-venous thromboembolism score. Crit Care 2020).

The authors cite the HEMORR2HAGES score, but this score not seems adaptable to a critically ill patient population. The “Bleeding score” appears more appropriate (Decousus H, et al. Chest 2011).

Management of coagulopathies

The paragraph does not clearly introduce the topic. The different coagulopathies that will be explained later are not well defined here.

Laboratory and point-of-care assessment: the topic is too dispersive and ranges from standard laboratory tests to viscoelastic tests, then it moves on to aggregometry, confusing the reader without a clear message of clinical practice.

The authors tried to explain different types of coagulopathies in critically ill patients without focusing on the main aspects. Furthermore, they considered topics not inherent to the purpose of this review (i. e. Mechanical circulatory support devices; Role of plasma in the treatment of coagulopathy).

In conclusion, I would suggest to the authors to focus on the chosen topic (coagulopathies in critically ill patients or patient blood management in critically ill patients or thromboprophylaxis/anticoagulation in critically ill patients or reversal of anticoagulants in bleeding critically ill patient) and, carry out a systematic review or a simple narrative review of the literature.

Minor comments

Introduction

Line 64: About the guidelines on thromboprophylaxis (ref. 8-9), European Guidelines are missing (Afshari A, et al. EJA 2018).

Line 73: “In patients with elevated PTT for unknown reasons, further evaluation and testing is warrant before anticoagulation is considered”: What are the authors referring to? Antiphospholipid antibodies such as lupus anticoagulant? Please explain.

Line 77: pneumatic compression device for VTE prophylaxis: please refer to PREVENT study. NEJM 2019.

Management of coagulopathies

Line 96: “Any bleeding patients should be risk-stratified based on hemodynamic instability, timing of the last dose of anticoagulant agent, source of bleeding, and degree of blood loss….”: obvious considerations without bibliographical references.

Paragraph 2.3: the use of tranexamic acid and fibrinogen is only cited. I suggest to the authors to go into the argument with more details, basing on the literature.

Author Response

Response to Reviewer 1 Comments

Point 1:

Abstract: The abstract is confused and the topic that will be treated is not clear (patient blood management in critically ill patients or coagulopathies or anticoagulation or reversal?). The authors seem to address the topic on patient blood management (BPM) and they point out that the cornerstones of coagulation management are patient evaluation for individual thrombosis and bleeding risks, analysis of different coagulopathies and their treatment with a goal-directed therapy. However, the optimization of hemostasis is not explained in the following text.

Response 1: We thank this reviewer for the feedback and understand the reviewer’s quest to address each of the topics in more depth. However, the goal of our review was to give a comprehensive overview on both sides of coagulopathy. We feel that hemorrhagic and thrombotic incidences can be addressed in one review, because they are strongly linked as both may be present at the same time, both are due to imbalances in coagulation and each may also occur as a result of (overshooting) attempts to treat the other. Patient Blood Management incorporates every measure deemed to reduce bleeding by improving patient management, including patient risk evaluation to optimize hemostasis. The optimization of hemostasis is addressed in paragraph 3 and more information on fibrinogen and tranexamic acid was added at the reviewer’s suggestion.

Point 2:

Introduction: The reader would expect the introduction to the topic coagulopathies or BPM. However, the evaluation of thrombotic and haemorrhagic risk in critically ill patient is explained. If the authors decide to discuss about thrombotic risk (Padua score has been created for hospitalized medical patients), I would suggest referring also a new score for critically ill patients (see Viarasilpa T, et al. Prediction of symptomatic venous thromboembolism in critically ill patients: the ICU-venous thromboembolism score. Crit Care 2020).

Response 2: We thank this reviewer for the suggestions. We separated the section on “VTE and bleeding risk assessment tools” and added an introduction to improve the story. We furthermore implemented the suggested relevant citation and also changed to references for the recently published meta-analysis: Tran A, Fernando SM, Rochwerg B, Cook DJ, Crowther MA, Fowler RA, Alhazzani W, Siegal DM, Castellucci LA, Zarychanski R et al: Prognostic Factors Associated With Development of Venous Thromboembolism in Critically Ill Patients-A Systematic Review and Meta-Analysis. Critical care medicine 2021.

Point 3:

The authors cite the HEMORR2HAGES score, but this score not seems adaptable to a critically ill patient population. The “Bleeding score” appears more appropriate (Decousus H, et al. Chest 2011).

Response 3: We incorporated the suggested “Bleeding score” into the revised manuscript and downgraded the HEMORR2HAGES score.

Point 4:

Management of coagulopathies: The paragraph does not clearly introduce the topic. The different coagulopathies that will be explained later are not well defined here.

Laboratory and point-of-care assessment: the topic is too dispersive and ranges from standard laboratory tests to viscoelastic tests, then it moves on to aggregometry, confusing the reader without a clear message of clinical practice.

The authors tried to explain different types of coagulopathies in critically ill patients without focusing on the main aspects. Furthermore, they considered topics not inherent to the purpose of this review (i. e. Mechanical circulatory support devices; Role of plasma in the treatment of coagulopathy).

In conclusion, I would suggest to the authors to focus on the chosen topic (coagulopathies in critically ill patients or patient blood management in critically ill patients or thromboprophylaxis/anticoagulation in critically ill patients or reversal of anticoagulants in bleeding critically ill patient) and, carry out a systematic review or a simple narrative review of the literature.

Response 4: We thank this reviewer for the feedback and understand the quest to address each of the topics in more depth. However, the goal of our review was to give a comprehensive overview on both sides of coagulopathy with a special focus on the bleeding situations an intensivist will be confronted with. We feel that hemorrhagic and thrombotic incidences are strongly linked as both may be due to imbalances in coagulation and both may also be results of attempts to treat the other. We tried to streamline the paragraphs on “Management of coagulopathies” (now paragraph 3) by adding a clearer introduction and underlining the clinically most relevant messages regarding laboratory and POC assessment (being aware that the scope of this manuscript can only touch the most important aspects, referring the interested reader to other relevant literature for further reading). 

We are certain that coagulopathies resulting from mechanical circulatory support devices do indeed interest intensivists especially in times of our current global pandemic, necessitating ECMO-therapy in many cases. Also, the adequate transfusion of plasma is a common topic of debate in coagulation management and we are therefore convinced that it should be covered in this review.

Point 5:

Introduction:

Line 64: About the guidelines on thromboprophylaxis (ref. 8-9), European Guidelines are missing (Afshari A, et al. EJA 2018).

Line 73: “In patients with elevated PTT for unknown reasons, further evaluation and testing is warrant before anticoagulation is considered”: What are the authors referring to? Antiphospholipid antibodies such as lupus anticoagulant? Please explain.

Line 77: pneumatic compression device for VTE prophylaxis: please refer to PREVENT study. NEJM 2019.

Response 5: The prolonged PTT may be due to acquired or inherited causes such as von Willebrand syndrome or factor deficiencies. We added this information in line 75. We thank the reviewer for pointing out the two additional references and added them accordingly.

Point 6:

Management of coagulopathies

Line 96: “Any bleeding patients should be risk-stratified based on hemodynamic instability, timing of the last dose of anticoagulant agent, source of bleeding, and degree of blood loss….”: obvious considerations without bibliographical references.

Response 6: Another true point. We added a highly appropriate reference.

Dhakal P, Rayamajhi S, Verma V, Gundabolu K, Bhatt VR: Reversal of Anticoagulation and Management of Bleeding in Patients on Anticoagulants. Clinical and applied thrombosis/hemostasis : official journal of the International Academy of Clinical and Applied Thrombosis/Hemostasis 2017, 23(5):410-415.

Point 7:

Paragraph 2.3: the use of tranexamic acid and fibrinogen is only cited. I suggest to the authors to go into the argument with more details, basing on the literature.

Response 7: We thank the reviewer for this valuable input. To stress the importance of fibrinogen and tranexamic acid further, we added a section and appropriate references in paragraph 3.3.

Reviewer 2 Report

The authors describe a very interesting review coagulopathies in intensive care medicine. Coagulopathy in critically ill patients is common and of multifactorial origin. Coagulopathy-associated risk of bleeding and the use of allogeneic blood products are independent risk factors for morbidity and mortality.

The manuscript is well structured, but some parts are missing some important facts that authors should add.

Lines 87-93: Authors should correct an incorrectly cited reference 17. Bleeding management is very demanding in patients with bleeding disorders such as afibrinogenemia. This is a demanding management when it is necessary to recommend treatment to ensure a balance of hemostasis between the replacement of coagulation factor and thromboprophylaxis. Authors should add these facts to the manuscript and at the same time quote the manuscript in which it was stated. Semin Thromb Hemost. 2016 Sep;42(6):689-92. doi: 10.1055/s-0036-1585079.

Lines 122-128: Authors should state that role of global hemostasis assay for guiding the perioperative management v rare bleeding disorders. A case study on the use of rotary thromboelastometry describes these statements. Authors should cite this manuscript: Thromb Res. 2020 Apr;188:1-4. doi: 10.1016/j.thromres.2020.01.024.

Massive bleeding usually leads to critically low levels of clotting factors and coagulopathy. In addition to blood loss anddilution, acidosis causes further depletion of key clotting factors including fibrinogen. Fibrinogen therapy is an important component of a multimodal strategy for the treatment of coagulopathic bleeding. It is necessary to add a subchapter that will deal with fibrinogen. Authors should cite this manuscript:  Blood. 2015 Feb 26;125(9):1387-93. doi: 10.1182/blood-2014-08-552000.

A small number of tables in the manuscript, the legend below Table 2 does not explain all the abbreviations. Authors could add some figures or other tables.

I have to say that with these 131 references. The authors cite most of the references have been published in the last 5 years.

Author Response

Response to Reviewer 2 Comments

The authors describe a very interesting review coagulopathies in intensive care medicine. Coagulopathy in critically ill patients is common and of multifactorial origin. Coagulopathy-associated risk of bleeding and the use of allogeneic blood products are independent risk factors for morbidity and mortality.

The manuscript is well structured, but some parts are missing some important facts that authors should add.

Point 1:

Lines 87-93: Authors should correct an incorrectly cited reference 17. Bleeding management is very demanding in patients with bleeding disorders such as afibrinogenemia. This is a demanding management when it is necessary to recommend treatment to ensure a balance of hemostasis between the replacement of coagulation factor and thromboprophylaxis. Authors should add these facts to the manuscript and at the same time quote the manuscript in which it was stated. Semin Thromb Hemost. 2016 Sep;42(6):689-92. doi: 10.1055/s-0036-1585079.

Response 1: We corrected the reference. We value the input regarding afibrinogenemia. However, our article specifically does not cover congenital coagulopathies as this would lead too far. The importance of fibrinogen is stressed additionally in a further passage in paragraph 3.3 and we hope that the manuscript now covers the non-congenital coagulopathies adequately.

Point 2:

Lines 122-128: Authors should state that role of global hemostasis assay for guiding the perioperative management v rare bleeding disorders. A case study on the use of rotary thromboelastometry describes these statements. Authors should cite this manuscript: Thromb Res. 2020 Apr;188:1-4. doi: 10.1016/j.thromres.2020.01.024.

Response 2: The suggested reference refers to congenital coagulopathies, which we actively decided to omit as the theme would go beyond the already wide scope of our article. Instead, we added more information on the applicability of routine coagulation tests such as PT and aPTT primarily being developed to monitor anticoagulants and not for predicting the bleeding or thrombotic risk.

Point 3:

Massive bleeding usually leads to critically low levels of clotting factors and coagulopathy. In addition to blood loss and dilution, acidosis causes further depletion of key clotting factors including fibrinogen. Fibrinogen therapy is an important component of a multimodal strategy for the treatment of coagulopathic bleeding. It is necessary to add a subchapter that will deal with fibrinogen. Authors should cite this manuscript:  Blood. 2015 Feb 26;125(9):1387-93. doi: 10.1182/blood-2014-08-552000.

Response 3: We thank the reviewer for this valuable input. To stress the importance of fibrinogen further, we added a section in paragraph 3.3. We added the suggested citations and hope that the editor concurs with the rather high number of references.

Point 4:

A small number of tables in the manuscript, the legend below Table 2 does not explain all the abbreviations. Authors could add some figures or other tables.

Response 4: We thank the reviewer for pointing this out and we added the missing abbreviations. Furthermore, we thoroughly discussed the possibility of adding further figures or tables. However, we were uncertain and in disagreement as to which information would benefit from a presentation in form of a figure or table. We feel that the two existing tables already lighten up the reading experience and are valuable as a conclusive reference. We hope that the reviewer agrees and are otherwise open for further suggestions.

Round 2

Reviewer 1 Report

The manuscript has improved markedly.
The authors replied clearly to my comments.
I have no other requests.

Reviewer 2 Report

The presented manuscript has been corrected in response to the suggestions. The authors have followed the recommendations of the reviewer. After the revision, the provided data and addition of the results became more clear. I would like to thank the authors for resubmitting the manuscript and explaining the obscure points from the previous version.